# A Comparison of Two Versions of the CRISPR-Sirius System for the Live-Cell Visualization of the Borders of Topologically Associating Domains

**DOI:** 10.3390/cells13171440

**Published:** 2024-08-27

**Authors:** Vladimir S. Viushkov, Nikolai A. Lomov, Mikhail A. Rubtsov

**Affiliations:** 1Department of Molecular Biology, Faculty of Biology, Lomonosov Moscow State University, Moscow 119234, Russia; lomov@mail.bio.msu.ru (N.A.L.); ma_rubtsov@mail.ru (M.A.R.); 2Department of Biochemistry, Center for Industrial Technologies and Entrepreneurship, I.M. Sechenov First Moscow State Medical University (Sechenov University), Moscow 119435, Russia

**Keywords:** CRISPR-imaging, CRISPR-Sirius, live-cell microscopy, 4D genome, chromatin visualization, topologically associating domains (TADs), cohesin

## Abstract

In recent years, various technologies have emerged for the imaging of chromatin loci in living cells via catalytically inactive Cas9 (dCas9). These technologies facilitate a deeper understanding of the mechanisms behind the chromatin dynamics and provide valuable kinetic data that could not have previously been obtained via FISH applied to fixed cells. However, such technologies are relatively complicated, as they involve the expression of several chimeric proteins as well as sgRNAs targeting the visualized loci, a process that entails many technical subtleties. Therefore, the effectiveness in visualizing a specific target locus may be quite low. In this study, we directly compared two versions of a previously published CRISPR-Sirius method based on the use of sgRNAs containing eight MS2 or PP7 stem loops and the expression of MCP or PCP fused to fluorescent proteins. We assessed the visualization efficiency for several unique genomic loci by comparing the two approaches in delivering sgRNA genes (transient transfection and lentiviral transduction), as well as two CRISPR-Sirius versions (with PCP and with MCP). The efficiency of visualization varied among the loci, and not all loci could be visualized. However, the MCP-sfGFP version provided more efficient visualization in terms of the number of cells with signals than PCP-sfGFP for all tested loci. We also showed that lentiviral transduction was more efficient in locus imaging than transient transfection for both CRISPR-Sirius systems. Most of the target loci in our study were located at the borders of topologically associating domains, and we defined a set of TAD borders that could be effectively visualized using the MCP-sfGFP version of the CRISPR-Sirius system. Altogether, our study validates the use of the CRISPR-Sirius technology for live-cell visualization and highlights various technical details that should be considered when using this method.

## 1. Introduction

The technology for the visualization of individual chromatin loci in living cells is actively being developed. Much of the new technology is based on variants of the CRISPR-Cas system using catalytically inactive Cas9 (dCas9) (see recent reviews, [1,2,3,4,5,6,7]). The use of such techniques presents new pathways for the comprehensive study of chromatin dynamics, including chromatin movement accompanying replication, transcription and repair; the interaction of genes with nuclear subcompartments; and the mechanisms behind the formation of chromatin loops, topologically associating domains, and compartments [2,6,8]. Such methods can be used to “revive” data that were previously obtained through static methods such as FISH and immunostaining and thus provide valuable information concerning the kinetics of molecular processes occurring in individual cells. The development of imaging technologies based on the CRISPR-Cas system is an example of the application of synthetic biology: various modules are added to the basic framework of dCas9 and sgRNA to allow for imaging, enhancing the brightness of the signal and increasing the signal-to-noise ratio. The customization of the dCas9-sgRNA pair for imaging purposes is realized by two main approaches: adding fluorescent proteins, protein fragments, or peptide tags to dCas9 [9,10,11,12,13,14,15] or integrating protein-binding repeat sequences (e.g., MS2 or PP7 repeats) to the sgRNA scaffold that can be recognized by the corresponding proteins (e.g., MCP or PCP), which are then fused to the fluorescent proteins [15,16,17,18,19,20,21,22].

One such technology is known as CRISPR-Sirius [22,23,24]. This method is based on the use of sgRNAs with eight MS2 or PP7 stem loops integrated into an sgRNA tetraloop, combined with the cellular expression of fluorescence-tagged proteins that recognize such repeats. The MS2 and PP7 stem loops are regions of RNA bacteriophages that are recognized by the capsid proteins MCP and PCP of the phages MS2 and PP7, respectively (Figure 1). In the original work [22], the authors visualized two loci simultaneously by using an MCP-HaloTag for one locus and a PCP-sfGFP for the other locus. Octet arrays of repeats are not identical copies but rather may contain point mutations that, as the authors demonstrated, do not interfere with the binding of the PCP or MCP proteins but do increase the stability of sgRNAs with such repeats, thereby increasing the signal brightness. However, imaging remains possible only for loci that contain an array of short repeats (at least 20) located next to each other that can serve as a target for the sgRNA. This narrows the range of genomic loci that can be visualized using the CRISPR-Sirius technology. However, the number of loci containing clusters of locus-specific repeats to which dCas9 can potentially be targeted is still quite large, being approximately 1200 in the human genome [22]. CRISPR-Sirius has now been utilized in several published studies. The first article proposed the original concept [22]; the second described a study of the cell-cycle-dependent dynamics of several genomic loci via this technology [23], and the third examined the influence of the radial position, cell cycle stage, and transcriptional status on the spatial dynamics of a set of genomic targets [24].

Although MCP and PCP proteins are often assumed to be equivalent for imaging in live cells, the efficiency of locus imaging using the two proteins may differ. In this study, we compared the efficiency when imaging loci with MCP and PCP versions of the CRISPR-Sirius system. We visualized a set of target genomic loci and compared two methods of delivering the visualizing sgRNAs (transient transfection and transduction), in each case assessing the efficiency of visualization as the proportion of cells that contained nuclear signals (foci). We also evaluated the signal-to-background ratio for the two versions. To characterize the mechanisms underlying the differences in the efficiency of the MCP and PCP CRISPR-Sirius systems, we assessed the expression of sgRNAs and stem-loop-binding proteins for these two systems. 

Most of the target loci in our study were located at the borders of topologically associating domains (TADs)—sub-megabase-scale sequences that interact preferentially with themselves rather than with other regions of the genome [25]. TADs were discovered as a statistical phenomenon by a population Hi-C approach [26,27,28], and their existence was subsequently demonstrated in individual cells [29,30]. TADs in mammalian cells are thought to be formed by cohesin extrusion, and the boundaries of TADs correspond to sites where extrusion has stopped (for example, at CTCF sites) [31,32]. This hypothesis has been confirmed using Hi-C on cells in which cohesin [33,34,35], CTCF [35,36], and the cohesin unloader factor WAPL were depleted [33,35,37]. In addition, this model has been confirmed by an Oligopaint FISH analysis of TAD-like structures in individual cells [30]. However, Hi-C and FISH are only applied to non-living, fixed cells, and therefore do not allow the analysis of the kinetics of TAD formation or the study of this process in individual living cells. In several recently published studies, the authors managed to visualize the dynamics of the TAD boundaries in living cells [8,38,39]. In two of these studies, the methods required the integration of artificial sequences into target loci to which the corresponding proteins could bind: these were the ANCHOR/ParBS and Repressor/Operator systems [38,39]. Despite the essential data obtained in these works, the influence of the integrated sequences on the dynamics of the visualized loci could not be excluded. In a third study, Clow et al. used Casilio, a CRISPR-based approach similar to CRISPR-Sirius, to examine the arrangement of chromatin loop anchors [8]. CRISPR imaging alleviates the need for the integration of artificial sequences into the genome and is therefore relatively easy to implement. Complementary to this work, we attempted to assess the applicability of the CRIPSR-Sirius technology in both PCP and MCP versions to visualize several TAD boundaries. We were able to visualize several such boundaries, and the MCP system was more efficient for this purpose.

Our analysis of the CRISPR-Sirius system will allow researchers to realize its applicability and understand the possible challenges of using CRISPR-Sirius and similar imaging technologies. Our selected set of TAD boundary loci that we visualized using the CRISPR-Sirius technology can be used in future research to study the mobility of these boundaries and the mechanisms of genome folding. 

## 2. Materials and Methods

### 2.1. Plasmid Construction

sgRNA expression vectors with eight MS2 or PP7 repeats [22] were ordered from Addgene (121939 and 121940, respectively). To insert recognition sequences for the sgRNAs, the vectors were treated with the BstV2I restriction enzyme and the phosphatase FastAP (Thermo Fisher Scientific Baltics UAB, Vilnius, Lithuania). Then, the sgRNA chains were ligated onto the treated vector in the form of kinated and annealed DNA oligonucleotides with sticky 5′-overhangs (ACCG for the direct chain and AAAC for the reverse chain). The sgRNA recognition sequences and the genomic coordinates of the target loci are listed in Tables 2 and 4 in the Section 3. The sequence for IDR3 was taken from previous reports [22,23]; the sequences for the other sgRNAs were selected using the CRISPRbar online tool (http://genome.ucf.edu/CRISPRbar/, accessed on 20 May 2022), developed by the authors of the original CRISPR-Sirius article to select locus-specific short tandem repeats [22].

To replace the HSA marker with the puromycin resistance ORF, the dCas9 plasmid [22] from Addgene (121936) was treated with BamHI and XbaI restriction enzymes, and a fragment containing most of the vector and the dCas9 gene was isolated from the gel. This fragment was then ligated in-frame to the BamHI and XbaI digested PCR product containing the puromycin resistance open reading frame (ORF) and the T2A peptide amplified from the Puro-Cas9 donor plasmid (Addgene 58409). The resulting vector contained the T2A-Puro fragment instead of P2A-HSA. The HaloTag ORF in a plasmid for the expression of MCP (Addgene 121937) was replaced at the BamHI and XhoI sites by the sfGFP ORF. The open reading frame encoding sfGFP was amplified from a PCP-sfGFP-encoding plasmid (Addgene 121938). The obtained plasmids were sequenced to verify the inserts. Plasmids encoding dCas9-T2A-Puro, PCP-sfGFP, and MCP-sfGFP, as well as visualizing sgRNA-encoding plasmids, were self-inactivating transfer vectors for the assembly of second-generation lentiviruses and were used to produce lentiviral particles. The sequences of the primers used for molecular cloning are listed in Table 1. Annotated sequences of the obtained plasmids in .gbk format are provided in the Appendix A.

### 2.2. Cell Culture and Transient Transfection

HCT116 cells were cultured in DMEM with 10% fetal bovine serum (Capricorn Scientific GmbH, Ebsdorfergrund, Germany) at 37 °C in an atmosphere of 5% CO_2_. The transient transfection of HCT116 cells was performed with Lipofectamine 3000 (Thermo Fisher Scientific, Carlsbad, CA, USA). In this step, approximately 800,000 cells were seeded into the wells of a six-well plate in 2 mL of DMEM medium on the day before transfection. On the following day, 4 μg of one or several plasmids mixed with 6 μL of Lipofectamine 3000 and 6 μL of P3000 reagent (from the Lipofectamine kit) in 250 μL of Opti-MEM medium (Gibco, Grand Island, NY, USA) was added to the wells. The day after transfection, the cells were seeded onto 35 mm glass-bottomed dishes and were examined by microscopy one day after seeding (i.e., 48 h after transfection). These transfection conditions resulted in transfection efficiency of 40% as assessed by the sfGFP-encoding plasmid 48 h after transfection (Appendix A).

### 2.3. Lentivirus Production and Cell Transduction

To obtain lentiviruses, HEK293T cells were transfected with 4 μg of the pCMV-VSV-G (Addgene 8454) and pCMV-dR8.2-dvpr (Addgene 8455) plasmids and a transfer plasmid containing the gene of interest (dCas9-T2A-Puro, MCP-sfGFP, PCP-sfGFP, visualizing sgRNAs). Transfection was performed with the Turbofect reagent (Thermo Fisher Scientific, Carlsbad, CA, USA), 10 μL of which was mixed with the plasmids in 400 μL of Opti-MEM medium (Gibco, Grand Island, NY, USA). The mixture was added to the cells seeded on the previous day (approximately 300,000 in 2 mL of DMEM) in one well of a six-well plate. Four days after transfection, the medium with lentiviral particles was collected from the transfected cells, and the obtained viral suspension was filtered through a 0.2 μm filter.

For transduction, 300,000–400,000 target cells were seeded into one well of a six-well plate in 2 mL of DMEM, and 100–150 μL of lentiviral suspension was immediately added, along with up to 8 μg/mL of Polybrene reagent (Sigma-Aldrich, St. Louis, MO, USA). Four days after transduction, the cells were subcultured into media containing the required antibiotic: either puromycin at a concentration of 0.2 μg/mL (after transduction with viruses containing the dCas9-T2A-Puro gene) or hygromycin at a concentration of 200 μg/mL (after transduction with viruses containing visualizing sgRNA genes with 8xMS2 or 8xPP7 repeats). Alternatively, the transduced cells were selected using a cell sorter (after transduction with viruses containing MCP-sfGFP or PCP-sfGFP genes). The selection of cells with antibiotics was conducted until complete cell death in the control, non-transduced cultures that were set up in parallel to the experimental cultures in media containing the antibiotics at the same concentrations.

### 2.4. FACS

The cell lines employed in the experiments were sorted according to the single fluorescent tag sfGFP using a 488 nm excitation laser. The detection filters were 505LP+515/20BP. Sorting was performed using a five-laser FACSAria SORP instrument (BD Biosciences, Franklin Lakes, NJ, USA) with an 85 μm nozzle and the corresponding system pressure parameters. The sorting gates were set for each experiment according to the non-fluorescent negative control cell lines. The cells were loaded for sorting in serum-free medium/PBS to avoid aggregation and subsequently sorted into the full medium. The intensity of the sfGFP fluorescence in HCT116_dCas9_MCP-sfGFP and HCT116_dCas9_PCP-sfGFP cells was analyzed in the same way but in flow cytometry mode.

### 2.5. Western Blot

For each sample preparation, 500,000 cells were detached with trypsin, washed with DPBS, and lysed in 50 μL of 1x SDS-PAGE sample buffer (1% SDS, 10% glycerol, 32 mM Tris–HCl pH 6.8, and 0.01% bromophenol blue) containing 1 μL of Proteinase inhibitor cocktail (Sigma-Aldrich, St. Louis, MO, USA), 2 mM PMSF, and 30 U of Benzonase Nuclease (Merck Millipore, Burlington, MA, USA) at 37 °C in a thermal shaker for 20 min. DTT was then added to the lysates to adjust the concentration to 50 mM. The lysates were incubated at 95 °C for 5 min, and 10 μL of the lysates was loaded into the wells of a 1 mm polyacrylamide gel (6% stacking gel, 12% resolving gel). Then, 10 μL of prestained protein ladder RAV-11 (Biolabmix, Novosibirsk, Russia) was added at the edges of the gel. After separation, the proteins were transferred to a Hybond-C extra nitrocellulose membrane (Amersham Biosciences, Little Chalfont, Buckinghamshire, UK) by wet transfer in Towbin buffer (25 mM Tris, 192 mM glycine) with 10% methanol for two hours at 80 V in a Mini Trans-Blot Cell (Bio-Rad, Hercules, CA, USA). After the transfer, the membrane was stained with Ponceau S and photographed (see below) for further normalization to the total protein. The Ponceau S was then washed twice with TBST buffer for five minutes, after which the membrane was blocked for one hour at room temperature in TBST buffer with 5% skim milk. The membrane was then incubated overnight at +4 °C in a solution of anti-GFP primary antibodies (AB011, Evrogen, Moscow, Russia) diluted 1:2500 in TBST with 1% BSA. The next day, the membrane was washed four times for five minutes each time with TBST, after which it was incubated for two hours at room temperature in a solution of HRP-conjugated secondary antibodies (ab205718, Abcam, Cambridge, UK) diluted 1:5000 in TBST buffer with 1% BSA. The membrane was washed three times for five minutes each with TBST and one more time with TBS, after which it was developed using a Clarity Western ECL Substrate Kit (Bio-Rad). The membrane was photographed on a ChemiDoc Touch Imaging System (Bio-Rad) at an exposure level that did not lead to the appearance of oversaturated pixels. Images were processed in the Fiji software version 1.53t for background subtraction and the quantitative densitometric analysis of the target MCP-sfGFP and PCP-sfGFP protein bands. The Ponceau-S-stained image was processed in the Fiji software version 1.53t in the same manner for total protein normalization. No brightness correction was performed on the images. Raw membrane images (before background subtraction) are provided in the Appendix A.

### 2.6. Real-Time PCR

Total RNA was isolated from cells expressing dCas9, MCP-sfGFP, or PCP-sfGFP and the corresponding sgRNAs using a Ruplus-250 kit (Biolabmix) on spin columns according to the manufacturer’s protocol. The quality of the isolated RNA was verified by electrophoresis in agarose gels. Before cDNA synthesis, 5 μg of each RNA sample was treated with 1 U of DNase I (Thermo FS) at 37 °C for 30 min, and then EDTA was added to bring the concentration to 5 mM, and the DNase was inactivated by heating the samples to 65 °C for 10 min. cDNA synthesis was carried out using an RNAscribe RT kit (Biolabmix) according to the manufacturer’s protocol. The reaction was performed with 1 μg of RNA treated with DNase in a 20 μL reaction volume. Quantitative PCR was performed using 2x BioMaster UDG HS-qPCR SYBR Blue master mix (Biolabmix) containing SYBR Green I intercalating dye. The primers used for the sgRNA PCR were GTCCGAAAGGTGGCAAACAC and GACTCGGTGCCACTTTTTCA. These primers recognize regions identical in both the 8xMS2 and 8xPP7 variants of sgRNAs, and the sequences of the PCR products are also the same (except for the single-nucleotide difference T(MS2)—C(PP7) in the middle PCR product). The *GAPDH* gene was used as a calibrator gene, and its cDNA was amplified using the primers CAAGGTCATCCATGACAACTTTG and GTCCACCACCCTGTTGCTGTAG. Then, 0.5 μL of the corresponding cDNA sample was added to a 15 μL PCR reaction. The concentration of the primers in the reaction was 250 nM. PCR was performed on the CFX Connect Real-Time PCR Detection System (Bio-Rad). For the primer pairs used, the optimal annealing temperature was initially determined to be 63 °C. The PCR program was as follows: 50 °C 2 min–95 °C 5 min–(95 °C 15 s–63 °C 20 s–72 °C 40 s—plate read) × 36—melting curve 65–95 °C at 0.5 °C/5 s. The amplification efficiency of this protocol was determined using the standard curve method, and the values were 2.01 for the sgRNA pair and 1.88 for the GAPDH pair. The relative expression levels were calculated using the Pfaffl method [40]. The specificity of the PCR products was determined by a melting curve and electrophoretic analyses. PCR products were not detected for control reactions using isolated RNA treated with DNase but without cDNA synthesis.

### 2.7. Live-Cell Microscopy

The day before the microscopy, the cells were seeded onto 35 mm glass-bottomed dishes containing standard culture medium. Confocal microscopy was used to image the cells throughout this study. The images were acquired with an Olympus Fluoview FV3000 microscope equipped with a 40x UplanXApo dry objective and a stage-top incubator with an STX Temp and Flow module (Tokai Hit, Fujinomiya, Shizuoka, Japan). The sfGFP fluorescence was excited using a Coherent OBIS488 laser. The Olympus FV31S-SW software version 2.6 was used to control the microscope. Microscopy was performed at 37 °C in a humidified atmosphere with 5% CO_2_. In each case, z-stacks of several random fields of view were taken to ensure that the total number of cells per experiment was at least 100 and the total number of control cells was at least 200. Image analysis was performed with the Fiji software version 1.53t [41]. The images shown throughout the article are z-projections constructed via the maximum intensity method. To estimate the signal-to-background ratio, the Plot Profile tool of the Fiji software version 1.53t was used. The signal-to-background ratio was estimated only for stably transduced cells since the number of cells with signals in this case was higher than for transiently transfected cells. 

### 2.8. Graphical Representation of Hi-C and ChIP-Seq Data and Selection of Appropriate TAD Boundaries for Visualization

We used the Hi-C data for untreated HCT116 cells from Rao et al. [34] deposited at https://data.4dnucleome.org/ (experiment set 4DNES3QAGOZZ, accessed on 20 May 2022). Hi-C heatmaps for the hg38 genome assembly were visualized using the built-in visualization tool on the same portal; the maps were visualized at a 5 kbp resolution. Visualization-appropriate TAD boundaries were manually selected by matching the Hi-C maps to the coordinates of tandem repeats suitable for CRISPR-Sirius imaging. The coordinates of such loci with repeats were taken from the CRISPRbar server [22] (http://genome.ucf.edu/CRISPRbar/, accessed on 20 May 2022) with the following default parameters: total number of on-targets for each primer = 20, off-target percentage = 20, off-target density range = 50,000. For this matching, a Hi-C map for untreated HCT116 cells and a bed-file with repeat cluster coordinates were analyzed using the Juicebox software version 1.11.08 [42]. 

The ChIP-Seq profiles of the cohesin subunit RAD21 for the untreated HCT116 cells from Rao et al. [34] were downloaded from NCBI GEO (accession GSM2809609) and converted from the hg19 genome assembly to the hg38 genome assembly using CrossMap [43]. The visualization of the ChIP-Seq profiles was carried out using the IGV software version 2.11.3 [44]. 

To determine the number of clusters with repeats suitable for visualization at a given distance from the TAD boundaries, we wrote a custom script that used as input the coordinates of the TAD boundaries of HCT116 cells (file 4DNFIBKY9EG9 from data.4dnucleome.org, data from Rao et al. [34]), as well as the coordinates of clusters with tandem repeats suitable for visualization using the CRISPR-Sirius technology (from genome.ucf.edu/CRISPRbar; the parameters were as specified above), and returned the number of such clusters at a given distance from the TAD boundary.

### 2.9. Epigenetic Data 

We used epigenetic tracks for the reference epigenome of the HCT116 culture from the ENCODE portal (https://www.encodeproject.org/reference-epigenomes/ENCSR361KMF/, accessed on 6 June 2024). We used the following tracks: ENCFF259PSA (ATAC-Seq), ENCFF787LMI (H3K27Ac), ENCFF717ZKL (H3K27me3), ENCFF024LGD (H3K36me3), ENCFF254TIW (H3K9me3), ENCFF239FXT (H3K4me1), and ENCFF649ZLF (H3K4me3). The listed tracks represented a fold change over the control for two isogenic replicates. We also used the PolyA-plus RNA-seq track (ENCFF758LNP, signal of unique reads) from the same HCT116 reference epigenome. For the ChromHMM18 data, we used the ENCFF290AGK file from the ENCODE portal. The Hi-C compartment types for the target loci were assigned according to the 4DNFIZHT1Y8P compartment track for HCT116 cells from https://data.4dnucleome.org/ (data from Rao et al. [34], accessed on 6 June 2024). All epigenetic tracks were visualized in the IGV software version 2.11.3 [44]. 

## 3. Results

### 3.1. Evaluation of the PCP Version of the CRISPR-Sirius System

We first obtained cells expressing the dCas9 protein. To simplify the selection of dCas9-expressing cells, we replaced the CD24 marker (HSA) in the original dCas9 expression vector [22] with the puromycin resistance ORF. This modification alleviated the need for special antibodies for HSA to select cells. We obtained lentiviral particles using this new transfer vector, transduced HCT116 cells, and selected dCas9-expressing cells on puromycin-containing media. We then transduced these cells with lentiviruses with PCP-sfGFP—the original protein of the CRISPR-Sirius system [22]—and selected cells expressing PCP-sfGFP by FACS. The resulting culture was designated HCT116_dCas9_PCP-sfGFP.

We chose two initial genomic loci to evaluate the performance of the CRISPR-Sirius system. The first, IDR3, is a subtelomeric intergenic region on the short arm of chromosome 19 and was visualized in the original CRISPR-Sirius papers [22,23]. The second locus was a border of a topologically associating domain on chromosome 6 (located in the third intron of the TMEM242 gene) that we visualized here for the first time. This locus is designated “C6” throughout this paper. To visualize the IDR3 locus, a single sgRNA was used to target 45 repeats in a 2 kb region of the genome. To visualize the C6 locus, two sgRNAs were used, one (C6_g1) recognizing 22 repeats and the other (C6_g2) recognizing 39 repeats in a 4 kb region (Table 2).

We employed two approaches to deliver the visualizing sgRNA genes into HCT116_dCas9_PCP-sfGFP cells: transient transfection with Lipofectamine and transduction followed by selection on hygromycin. Transient transfection permitted the detection of one to two fluorescent signals (fluorescence foci) in the nuclei, but signal-containing cells were rarely observed: only 6% of such cells were found when using sgRNA to the IDR3 locus, and 3% of cells with signals were observed when using sgRNA to the C6 locus (Figure 2A). However, the proportion of cells with signals increased when transduction followed by selection was used instead of transfection: signals were observed in 26% of cells for the IDR3 locus and 20% of cells for the C6 locus (Figure 2B). The control cells contained no signals, as they did not express the visualizing sgRNAs (Figure 2C). The histogram in Figure 2D shows the distribution of nuclei by the number of foci in cells stably transduced with the C6 and IDR3 sgRNA genes. This distribution was consistent with the visualization of unique genomic loci in the cells. The signal-to-background ratio was 1.6 ± 0.1 for the IDR3 locus and 2.9 ± 0.3 for the C6 locus (shown are mean values ± standard errors of the mean).

We observed that almost all cells containing signals in the nuclei also showed discrete foci of fluorescence in the cytoplasm (Figure 2A,B, sfGFP channel). Notably, the same foci could be found by carefully examining the images in the original article (see [22], Figure 1f). However, in control cells that did not express sgRNAs, such cytoplasmic signals were absent (Figure 2C), suggesting that their formation requires the expression of a sgRNA. We believe that such signals do not undermine the applicability of the CRISPR-Sirius technology, as they are located in a different cell compartment and are difficult to confuse with signals in the nucleus. We also noticed that most nuclei showed the accumulation of PCP-sfGFP in the nucleolus, including the control cells without sgRNA expression (Figure 2A–C), a phenomenon that has also been described in other studies using the PCP protein [16,18]. In some cells, the visualized loci were close to the nucleolus and thus may have been hidden by the bright nucleolus. This may at least partly explain why there were cells with no signals or only one signal.

### 3.2. Evaluation of the MCP Version of the CRISPR-Sirius System

Similar to the PCP version described in the previous section, we assessed the performance of the version containing the MCP protein. To exclude the influence of the fluorescent protein on the comparison, we replaced the original HaloTag protein for MCP with sfGFP. A new MCP protein variant (MCP-sfGFP) was delivered into dCas9-expressing HCT116 cells by lentiviral transduction, and the resulting MCP-sfGFP-expressing cells were selected by FACS. The resulting culture was designated HCT116_dCas9_MCP-sfGFP. 

We tested the new version of the system on the same target loci (ICR3 and C6), again using two options to deliver imaging sgRNA genes: plasmid transfection and lentiviral transduction. As in the previous case, plasmid delivery resulted in relatively low visualization efficiency. The IDR3 locus could be visualized in 13% of the cells, while the C6 locus was visualized in 9% of the cells (Figure 3A). After stable transduction, the proportions of cells with signals increased markedly (Figure 3B). The IDR3 locus was visualized in 33% of the transduced cells, while the C6 locus was visualized in 52% of the cells. For the IDR3 locus, the cells were approximately equally likely to show one or two signals; for the C6 locus, the cells exhibited two signals approximately twice as often as one signal (Figure 3D). Signals were not detected in the nuclei of cells that lacked visualizing sgRNAs (Figure 3C). Therefore, the MCP-sfGFP version of the CRISPR-Sirius system showed a higher proportion of cells with signals than the original PCP-sfGFP version in all tests (Table 3). The signal-to-background ratio was 2.3 ± 0.1 for the IDR3 locus and 2.8 ± 0.3 for the C6 locus (mean values ± standard error of the mean). Thus, the ratio for IDR3 was higher in the MCP-sfGFP version than in the PCP-sfGFP version, and the ratio for the C6 locus was the same for both types of the CRISPR-Sirius system.

Notably, cytoplasmic fluorescent aggregates were significantly less pronounced when using the MCP version; these aggregates were rarer and paler compared to those in the system with PCP. Interestingly, in contrast to the pattern observed using PCP-sfGFP, MCP-sfGFP tended to avoid the nucleolus (see Figure 3A–C).

### 3.3. Evaluation of the Imaging Performance Using a Single Guide RNA per Locus

We used two sgRNAs to visualize the C6 locus, with the assumption that using two sgRNAs should improve the imaging efficiency compared to using a single sgRNA. Previous research on CRISPR imaging has shown that increasing the number of guide RNAs directed to a target locus can improve the efficiency of imaging [9,21]. To investigate this issue, we compared the imaging efficiency of the C6 locus using two sgRNAs (C6_sg1 + C6_sg2) with the imaging efficiency using the C6_sg1 and C6_sg2 guide RNAs separately. We found that using two sgRNAs in the MS2/MCP-sfGFP system slightly improved the imaging efficiency compared to using C6_sg1 sgRNA alone and did not provide any efficiency gain or loss in other cases (Figure 4A). Within the same type of CRISPR-Sirius system, sgRNAs C6_sg1 and C6_sg2 individually were equally efficient. Nevertheless, the results of the comparison of sgRNAs individually confirmed the previous conclusion for paired guide RNAs: in all cases, the MS2/MCP-sfGFP system was more efficient than the PP7/PCP-sfGFP system.

The signal-to-background ratio increased slightly when using paired sgRNAs compared to using the C6_sg1 guide RNA alone for both systems: the median ratio increased by approximately 1.5 times for the PP7/PCP-sfGFP system and by approximately 1.3 times for the MS2/MCP-sfGFP system (Figure 4B). However, the signal-to-background ratio did not differ significantly when using a pair of sgRNAs compared to using the C6_sg2 guide RNA alone (again for both versions of the system). This observation supported the finding that using two guide RNAs can, in some cases, achieve better visualization than using a single guide RNA. The distribution of cells according to the number of signals was similar when using guide RNAs individually or in combination (Figure 4C,D). Based on these data, we recommend using two sgRNAs targeting the same repeat cluster whenever possible, as this may improve the imaging efficiency and increase the signal-to-background ratio compared to using a single sgRNA.

### 3.4. Analysis of the Expression of sgRNAs and Stem-Loop-Binding Proteins in Two Versions of the CRISPR-Sirius System

To explore the reasons for the differences in the efficiency of the MS2/MCP and PP7/PCP CRISPR-Sirius systems, we compared the expression levels of the MCP-sfGFP and PCP-sfGFP proteins using Western blotting and flow cytometry and assessed the expression of sgRNAs with MS2 and PP7 repeats using quantitative PCR. 

The expression analysis of MCP-sfGFP and PCP-sfGFP performed in HCT116_dCas9_MCP-sfGFP and HCT116_dCas9_PCP-sfGFP cells, respectively, showed higher levels of the PCP-sfGFP protein both in the Western blots and by flow cytometry (Figure 5A,B). We did not detect degradation products of the MCP-sfGFP or PCP-sfGFP proteins in the Western blot analysis, indicating that both proteins were stable. At first glance, the results seem contradictory, since it was the version with MCP-sfGFP that appeared more efficient. However, the higher efficiency of this variant appears to be due to lower background nuclear fluorescence due to the lesser MCP-sfGFP expression, and hence a higher proportion of cells exhibited signals in the MCP-sfGFP version.

We also compared the expression levels of sgRNAs individually (the C6 and IDR3 loci) and when the two sgRNAs were used simultaneously (the C6 locus) in cells expressing dCas9 and the corresponding stem-loop-binding proteins (PCP-sfGFP or MCP-sfGFP). For the sgRNA PCR, we used primers that recognized regions identical in both the 8xMS2 and 8xPP7 variants of the sgRNAs, making it possible to use the same primer pair to analyze the expression of both types of guide RNA. In all cases, the sgRNAs were expressed, but their expression levels varied (Figure 5C–E). However, we found no significant correlation between the sgRNA expression level and the imaging efficiency (Figure 5F), suggesting that the level of sgRNA expression is not a factor significantly affecting the efficiency of imaging (at least for the sgRNAs that we used). Taken together, our results suggest that the greater imaging efficiency of the MS2/MCP type of CRISPR-Sirius system was most likely due to lower levels of MCP-sfGFP protein expression rather than differences in the sgRNA expression levels.

### 3.5. Expanding the Set of Target Loci—Visualizing the Boundaries of TADs

Having compared the systems using MCP and PCP proteins for two test loci (IDR3 and C6), we decided to try to visualize loci located within the boundaries of topologically associating domains. Studying the dynamics of these loci would serve to test and complement existing hypotheses concerning the mechanisms of formation of these chromatin structures. In addition to the C6 locus described above, located at the TAD boundary, we selected several additional TAD boundary loci containing tandem repeat clusters suitable for imaging: 6T1_L, 6T2_L, 6T2_L, and 6T2_R on Chromosome 6; 4T on Chromosome 4; 5T on Chromosome 5; and 22T on Chromosome 22 (Table 4, Figure 6A). The selected loci are adjacent to cohesin-binding sites (Appendix A). For each locus, we used two sgRNAs that recognized repeats in one or two adjacent clusters in the target locus. Lentiviral transduction was used to deliver the sgRNA genes, as this was more efficient than the transfection in our previous tests. We attempted to visualize these loci using both the PCP and MCP versions. Among the selected loci, only the C6 locus could be visualized via both versions of the system (see Figure 2A,B, Figure 3A,B, and Figure 6B). Another four loci (6T2_R, 4T, 5T, and 22T) were visualized only using the MCP version (Figure 6B), and the efficiency of visualization of these loci varied (Table 5). Taken together, the data support the greater utility of MCP-sfGFP compared to PCP-sfGFP for the imaging of chromatin loci in living cells, particularly the boundaries of TADs.

### 3.6. Analysis of the Dependence of the Visualization Efficiency on the Number of sgRNA Repeats in a Cluster and Epigenetic Factors

Although the MS2/MCP-sfGFP version of the CRISPR-Sirius system showed better performance, we still observed variations in the imaging efficiency for this version of the system (from 0 to 52%, Table 5). This result could be explained by, among other factors, different numbers of sgRNA recognition sites in clusters and different epigenetic contexts, as well as factors depending on the exact sequence of the sgRNA and the target DNA (for example, different levels of sgRNA binding to their recognition sequences). We decided to determine whether the observed efficiency of visualization was influenced by the number of sgRNA repeats and the epigenetic context of the set of repeat clusters that we studied. We note, however, that the diversity of the clusters that we studied was too low to draw general conclusions, and thus the analysis is provided mainly for descriptive purposes and possible future meta-analyses. The third factor—the exact sequences of the sgRNAs and target DNA—was beyond the scope of the present study due to the limited number of loci, excluding the possibility of drawing sequence-specific conclusions (see Section 4). 

We first tested whether there was a correlation between the number of repeats in a cluster and the efficiency of its visualization. In all loci except 6T2_L and 6T2_R, we visualized one cluster recognized by two sgRNAs whose sites were interspersed (Table 4, Appendix A). The 6T2_L and 6T2_R loci contained two separate clusters (Figure 6A, Table 4). To evaluate the efficiency of visualization of individual clusters in these loci, we obtained cell cultures separately expressing the sgRNAs 6T2_L_sg1, 6T2_L_sg2, 6T2_R_sg1, and 6T2_R_sg2. For this experiment, the corresponding sgRNA genes were delivered by lentiviral transduction into HCT116_dCas9_MCP-sfGFP cells as before. Among these four sgRNAs, only 6T2_R_sg1 had non-zero imaging efficiency (31%). With this refinement, there were 11 clusters in our sample that we attempted to visualize using CRISPR-Sirius (the MS2/MCP-sfGFP version) in this study (Table 6).

We found no correlation between the number of repeats for the sgRNA in a locus and the efficiency of visualization of this locus (Figure 7A, Spearman correlation coefficient = 0.46, *p*-value = 0.156). However, there is likely a minimum acceptable number of sgRNA repeats in the target locus: we failed to visualize three out of three loci with less than 30 sgRNA repeats. However, due to the small sample size, it cannot be unequivocally stated that 30 repeats is the lower limit of the number of repeats in a locus that can potentially be visualized by the CRISPR-Sirius technology under our conditions (*p*-value = 0.06, Fisher’s exact test). 

To determine whether the epigenetic context influenced the efficiency of locus visualization, we examined the ATAC-Seq profiles and also studied the chromatin types of the target loci using the ChromHMM18 chromatin state model, which is based on the abundance of six epigenetic marks (H3K4me3, H3K27Ac, H3K4me1, H3K36me3, H3K9me3, and H3K27me3) and classifies chromatin into 18 functional types [45,46]. In addition, using the HCT116 transcriptome data, we determined whether the target clusters resided in transcriptionally active chromatin (Appendix A, Table 6).

ATAC-Seq data indicate the degree of chromatin openness (accessibility for transposase) [47]. One may expect that the loci that could be visualized should contain peaks within or near the ATAC-Seq profile, i.e., be more open. Among the loci studied, only C6 contained a weak ATAC-Seq peak within itself (Appendix A). Another ATAC-Seq peak was approximately 500 bp from the C6 locus boundary. We note that the loci that could be visualized more often contained an ATAC-Seq peak up to 2000 bp from the repeat cluster boundary. We were able to visualize all five loci that contained an ATAC-Seq peak at distances of up to 2000 bp. At the same time, only one out of six loci that contained the ATAC-Seq peak more than 2000 bp from the border of the repeat cluster could be visualized. This difference was statistically significant (*p*-value = 0.015, Fisher’s exact test). The Spearman correlation coefficient for the visualization efficiency and the distance to the ATAC-Seq peak from the locus border was −0.73 (*p*-value = 0.008), indicating the existence of a weak correlation between the visualization efficiency and the proximity to the ATAC-Seq peak (Figure 7B). The existence of such a correlation points to a link between the visualization efficiency and the degree of openness (accessibility) of chromatin for proteins. However, this factor was not decisive, since, for loci with a similar proximity to the ATAC-Seq peak (approximately 1.4 kbp for IDR3, 4T, and 5T loci), the visualization efficiency differed significantly (33%, 16%, and 5%, respectively). When considering two factors simultaneously—the number of guide RNA repeats in a locus and the distance to the ATAC-Seq peak—there was a more pronounced pattern: all of the loci that we managed to visualize were concentrated in the upper left sector of the diagram (Figure 7C). Accordingly, we were able to visualize loci that were located near the ATAC-Seq peaks and that had a larger number of repeats.

The analysis of the chromatin types (ChromHMM18 model) to which the target loci belonged was inconclusive, since almost all target loci belonged to the quiescent chromatin type, i.e., they did not contain significant amounts of epigenetic marks characteristic of other chromatin types (Table 6, Appendix A). However, we note that the 6T2_R_sg1 cluster, which was visualized with relatively high efficiency (31%), belongs to the weakly repressed Polycomb-associated chromatin. This observation suggests that the fact that a locus belongs to repressed chromatin does not exclude its effective visualization using CRISPR-Sirius. We also note that although the C6 locus belongs to the quiescent chromatin according to the ChromHMM18 model, when analyzing the RNA-Seq data, this locus was weakly transcribed (the repeat cluster is located in the intron of the *TMEM242* gene). This may be why its visualization efficiency was the highest in our sample, but it is impossible to conclude on the statistical significance of this pattern due to the small sample size. The type of Hi-C compartment also does not seem to be a determining factor. Thus, the efficiency of visualization of loci within the A compartment varied from 0 to 52% (Table 6). Since only one of the target loci was in the B compartment (22T, visualized with low efficiency), we cannot be certain whether the location of a locus in the B compartment imposes strict limitations on the efficiency of its imaging by the CRISPR-Sirius technology.

## 4. Discussion

Despite the rapid development of technology for the live-cell visualization of genomic loci, many published reports are “proof of concept” studies that lack either a thorough analysis of the effectiveness of the described visualization technology or a description of the pitfalls that arise in using these approaches. In our experience, technical difficulties almost always arise. Therefore, many researchers who are new to using these methods risk experiencing low efficiency or applying an unsuitable approach to a specific cell type or a specific locus. In this work, we determined the efficiency of the CRISPR-Sirius imaging of several genomic repeat-containing loci using two approaches to delivering visualizing sgRNA genes: transient transfection and stable transduction. In all cases, lentiviral transduction resulted in a greater proportion of cells with signals than transient transfection, even though the efficiency of transient transfection was relatively high (~40%, Appendix A). However, even when using lentiviral transduction, the proportion of cells with signals in the population was far from 100%. This may be explained by the heterogeneity of the cell population. Lentiviral transduction was used to deliver dCas9 genes as well as fluorescent protein genes (PCP-sfGFP and MCP-sfGFP), and thus the integration of these genes occurred in random regions of the genome with different epigenetic contexts, a factor that may have led to variations in the expression levels in different cells. In addition, the sgRNA expression levels will differ among cells. A certain optimal level of expression of the dCas9, PCP-sfGFP (or MCP-sfGFP), and visualizing sgRNA genes is likely required to generate signals, a level that was not achieved in all cells of the population. However, the percentage of cells produced through lentiviral transduction was sufficient to quickly identify the desired cells using microscopy: in each 70 × 70-μm field of view, there was at least one cell with signals in the nucleus. The lower percentage of cells produced by transient transfection can be explained by the fact that the imaging efficiency relies on the efficiency of cell transfection. 

A direct comparison of the two versions of the CRISPR-Sirius system using the PCP-sfGFP or MCP-sfGFP proteins revealed that the latter provided more efficient visualization in terms of the proportion of cells with signals for all tested loci (Table 3 and Table 5). The signal-to-background ratio was similar for both systems. Therefore, we recommend using sfGFP in combination with MCP instead of PCP for single-locus visualization in living cells. The conclusion concerning the greater efficiency of the MS2/MCP-sfGFP system remained valid when using one or a pair of guide RNAs per locus, as shown in the C6 locus example (Figure 4). We found that the two types of CRISPR-Sirius system differed significantly in the expression of the stem-loop-binding protein. Thus, MCP-sfGFP was expressed at a lower level than PCP-sfGFP. This may at least partly explain the difference in efficiency between the two versions: more cells exhibited signals using the MCP-sfGFP version, signals that would be too weak to be visualized by the PCP-sfGFP version. The level of sgRNA expression was not correlated with the visualization efficiency, indicating that this was not a factor in the case of our set of tested sgRNAs. However, there should be a threshold for the sgRNA expression level below which visualization will be unachievable. 

In some cells, we observed duplicated loci after replication (see Figure 3B as an example); replicated loci were observed for both versions of the CRISPR-Sirius system and both the IDR3 and C6 target loci. This observation provided evidence that the detected signals were not artifactual aggregates but rather corresponded to visualized unique chromatin loci. 

We observed fluorescent aggregate artifacts in the cell cytoplasm when using the PCP-sfGFP protein (Figure 2A–C and Figure 1f in [22]). The formation of aggregates required the expression of visualizing sgRNAs, as no such aggregates were seen in the control cells that lacked sgRNAs (Figure 2C and Figure 3C). The nature of such aggregates is interesting, but we believe that their location in the cytoplasm makes them visually distinct from the target signals in the nucleus. Notably, such aggregates were less common when using the MCP-sfGFP version of the CRISPR-Sirius system.

We employed the CRISPR-Sirius system to visualize the boundaries of topologically associating domains, and we were able to visualize several TAD boundaries. However, the version with PCP-sfGFP appeared to be poorly applicable to this task, again confirming our conclusions concerning the greater applicability of MCP-sfGFP in visualizing chromatin loci. However, even for the MCP-sfGFP version, the visualization efficiency varied from 3 to 52%, and some loci could not be visualized. As expected, the visualization efficiency depends on many factors, and we cannot unambiguously predict this from individual factors such as the number of sgRNA repeats in a cluster, the abundance of epigenetic marks in the target chromatin, the type of chromatin compartment, the transcriptional activity, or the degree of chromatin accessibility. Of these factors, only the degree of chromatin accessibility (proximity to ATAC-Seq peaks) was weakly correlated with the visualization efficiency. These observations highlight the need for the experimental validation of the selected sgRNAs. Since this can be a time-consuming and labor-intensive process, as a reasonable sgRNA selection strategy, we can recommend the use of several sgRNAs per target locus and also the selection of sgRNAs targeting more open, transcriptionally active chromatin. It is also worth noting that the precise sequence of the sgRNA, as well as the nucleotide context of the sgRNA recognition site, can have a significant effect on the visualization efficiency [48]. Our sample of sgRNAs was too small to reveal such patterns. In our study, as well as in other studies utilizing the CRISPR-Sirius technology [22,23,24] or other chromatin imaging methods [19,49], truncated guide sequences were used, with the size of the recognition region of all sgRNAs consisting of 12 nucleotides (except for the guide RNA C6_sg2, which comprised 15 nucleotides). The application of existing predictive models [48,50] to our set of guide RNAs is therefore impossible since they do not accept such short guide RNAs as input and are designed to predict the probability of certain genomic editing events and not the efficiency of imaging or at least the efficiency of binding dCas9 to repeats in the target DNA. Although, on average, shorter guide sequences are less efficient in the context of genome editing, practically disappearing at a length of less than 16 nucleotides [51,52,53,54,55,56], dCas9 binding to target DNA does not cease with such short guide RNAs, a situation that allows the use of short guide RNAs for the imaging and regulation of gene expression by dCas9-based technologies [53,54]. In the case of CRISPR-Sirius, the sgRNAs are directed to repeat clusters, thereby increasing the total number of imaging complexes. The need to use shortened guide RNAs is dictated by the greater prevalence of clusters of shorter repeats in the genome. However, to date, there has been no comparison of the visualization efficiency using CRISPR imaging using different guide RNA lengths (and, as a consequence, different ratios of the copy number of the guide RNA recognition sites to its length). Such data would allow the more accurate characterization of the tradeoff between the length of a guide RNA and the copy number of its recognition sites, which would make the choice of suitable guide RNAs more rational. Furthermore, significant advances in CRISPR imaging could be achieved by developing algorithms that predict the imaging performance based on the guide RNA sequence. 

We note that the number of TAD boundaries that can potentially be visualized using the CRISPR-Sirius technology exceeds the eight boundaries that we attempted to visualize in our study. Figure 8 shows the number of locus-specific repeat clusters suitable for visualization (as predicted by the CRISPRbar service [22]) as a function of the distance to the TAD boundaries in HCT116 cells. The number of these clusters increases linearly with the allowed distance to the TAD boundary. The choice of a biologically relevant distance from the TAD boundary is determined by the context of the study, but, even at a fairly small distance in the scope of optical microscopy—10,000 kb—the number of potentially visualized clusters was 90, a number that is high enough to study TAD borders in different types of chromatin.

## 5. Conclusions

In this study, we compared the efficiency of two versions of the CRISPR-Sirius system, PP7/PCP-sfGFP and MS2/MCP-sfGFP, and found that the latter was more efficient for the studied guide RNAs. The higher level of efficiency of this version may be due to the observed lower expression of MCP-sfGFP than PCP-sfGFP, rendering the absolute value of the background fluorescence lower in the version with MCP-sfGFP. However, there was no correlation between the level of sgRNA expression and the visualization efficiency. We also did not find a correlation between the number of sgRNA repeats in the target locus and the efficiency of visualization, but there was a weak positive correlation regarding the proximity of the target locus to the nearest ATAC-Seq peak. We also showed that lentiviral transduction more efficiently delivered the visualizing sgRNA genes in the CRISPR-Sirius technology than transient transfection. Our study highlights the need to experimentally validate the visualization of target loci using the CRISPR-Sirius technology and demonstrates that not all loci can, in principle, be visualized. Most of the loci that we studied were boundaries of topologically associated domains, and thus we compiled a list of TAD boundaries that could be visualized and studied using the CRISPR-Sirius technology in future research.

## Figures and Tables

**Figure 1 cells-13-01440-f001:**
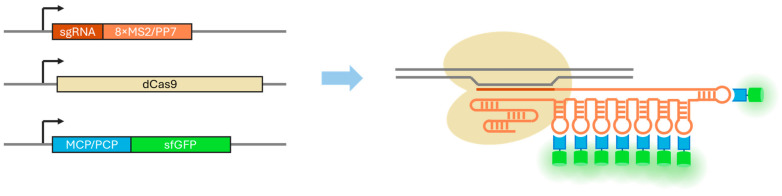
The principle of the CRISPR-Sirius visualization technology. The genes for dCas9, sgRNA with eight MS2 or PP7 stem loops, and a stem-loop-binding protein (MCP or PCP) fused to a fluorescent protein (e.g., sfGFP) are expressed in cells. The complex consisting of dCas9, sgRNA-8xMS2/PP7, and PCP/MCP-sfGFP binds to a target genomic region, thereby allowing its visualization by fluorescent microscopy. The technology was originally described in 2018 by Ma et al. [22].

**Figure 2 cells-13-01440-f002:**
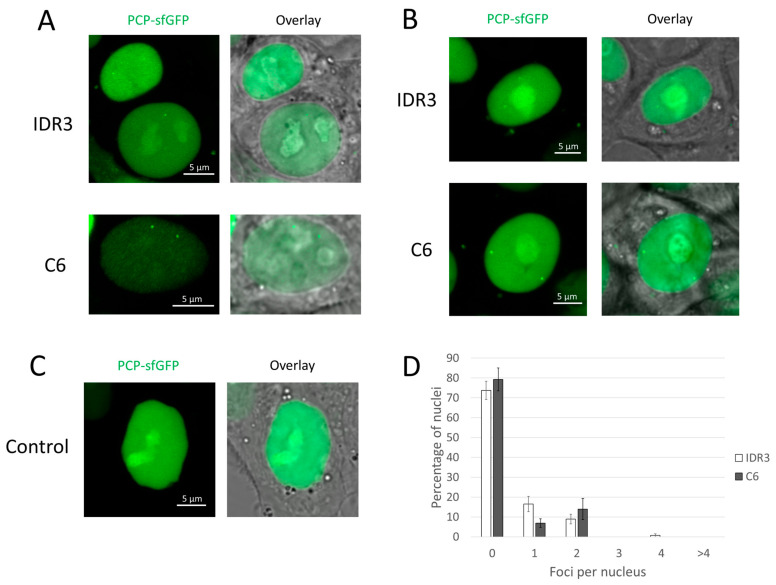
PCP-sfGFP version of the CRISPR-Sirius system. (**A**–**C**) Examples of microscope images of HCT116_dCas9_PCP-sfGFP cells. (**A**) Transient transfection with plasmids containing sgRNA genes for IDR3 or C6 (48 h after transfection); (**B**) stable lentiviral transduction with sgRNA genes for IDR3 or C6; (**C**) control cells that did not express sgRNAs. In each case, (**A**–**C**) show two images: in the sfGFP channel and an overlay of a fluorescent image on a corresponding bright field image. (**D**) The distribution of the number of observed foci in the nuclei of transduced cells with sgRNAs to IDR3 or C6. Mean values ± SEM after averaging by 10 fields of view for IDR3 (total 124 nuclei) and eight fields of view for C6 (a total 102 nuclei) are shown. No signals were visualized in any of the 257 imaged nuclei in control cells (a total of 17 fields of view).

**Figure 3 cells-13-01440-f003:**
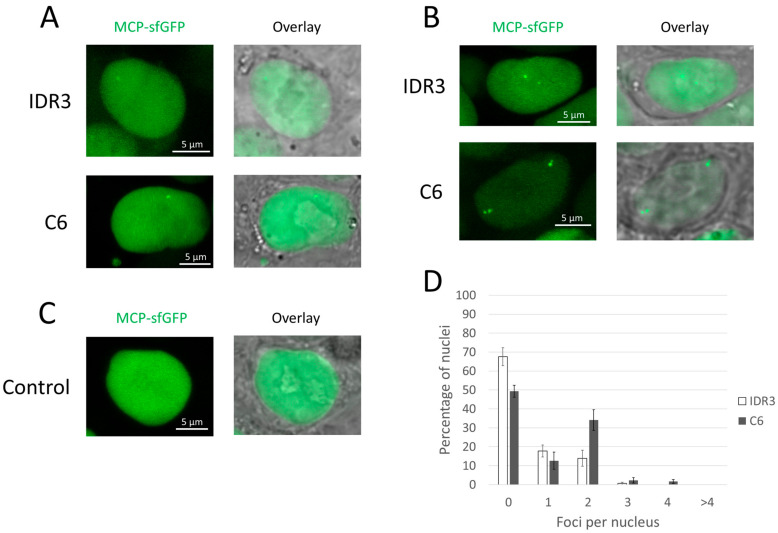
MCP-sfGFP version of the CRISPR-Sirius system. (**A**–**C**) Examples of microscope images of HCT116_dCas9_MCP-sfGFP cells. (**A**) Transient transfection with plasmids with sgRNA genes for IDR3 or C6 (48 h after transfection); (**B**) stable lentiviral transduction with sgRNA genes for IDR3 or C6; (**C**) control cells that did not express sgRNAs. In each case, (**A**–**C**) show two images: the sfGFP channel and an overlay of a fluorescent image on a corresponding bright field image. (**D**) The distribution of the number of observed foci in the nuclei of transduced cells with sgRNAs to IDR3 or C6. Mean values ± SEM after averaging by 5 fields of view for IDR3 (total 138 nuclei) and 6 fields of view for C6 (total 131 nuclei) are shown. No signals were visualized in any of the 191 imaged nuclei in control cells (a total of 11 fields of view).

**Figure 4 cells-13-01440-f004:**
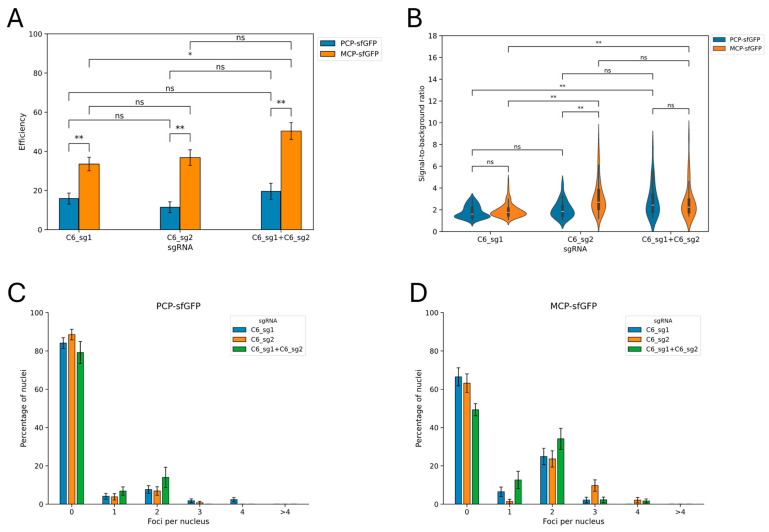
The results of the imaging efficiency analysis using one or two sgRNAs targeting the C6 locus. (**A**) Comparison of imaging efficiency using single or paired sgRNAs targeting the C6 locus. SgRNA genes were delivered by lentiviral transduction into cells expressing dCas9 and the corresponding type of stem-loop-binding proteins (HCT116_dCas9_PCP-sfGFP or HCT116_dCas9_PCP-sfGFP cells). The efficiency values are shown as the proportion of cells in which at least one signal was detected ± standard error for proportions; * *p* < 0.05, ** *p* < 0.005, ns: *p* > 0.05; two-proportion Z-test with Holm–Sidak correction for multiple comparisons. The total numbers of cells are 170, 185, 131, 144, 92, and 131 (the order corresponds to that on the graph). (**B**) Comparison of signal-to-background ratios using single or paired sgRNAs targeting the C6 locus. Violin plots correspond to kernel density estimation. The boxplots for corresponding samples are shown inside (white bar = median; box = interquartile range, whiskers = 1.5× interquartile range). ** *p* < 0.005, ns: *p* > 0.05, Mann–Whitney U-test with Holm–Sidak correction for multiple comparisons. The numbers of signals in each sample are 55, 116, 24, 124, 29, and 38 (the order corresponds to that on the graph). (**C**,**D**) The distribution of cells according to the number of visualized signals using the indicated guide RNAs, for two types of CRISPR-Sirius system: PP7/PCP-sfGFP (**C**) and MS2/MCP-sfGFP (**D**). The data values are the proportions of cells ± standard error for proportions.

**Figure 5 cells-13-01440-f005:**
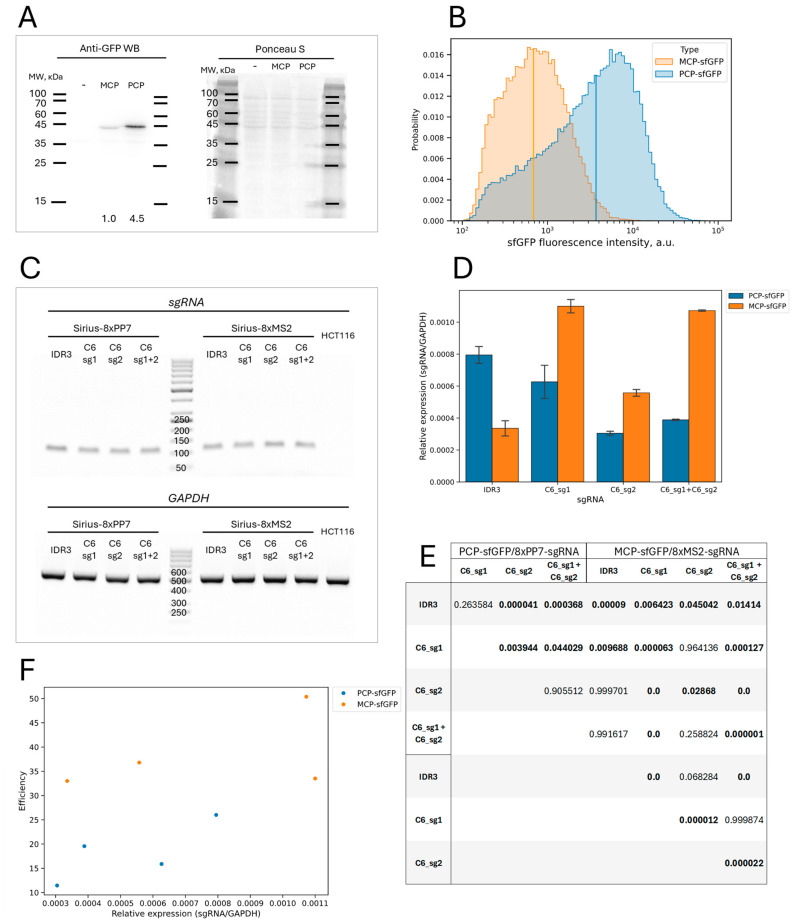
Analysis of the expression of sgRNAs and stem-loop-binding proteins in two CRISPR-Sirius systems. (**A**) Western blot analysis of MCP-sfGFP and PCP-sfGFP protein levels in HCT116_dCas9_MCP-sfGFP and HCT116_dCas9_PCP-sfGFP cells, respectively, using anti-GPF antibodies. The molecular weights of several marker bands are indicated. The left image corresponds to the ECL detection of the target proteins; the right image is the result of membrane staining with Ponceau S for total protein normalization. The numbers below the lanes correspond to the relative intensity of the target band after normalization to the total protein. Lane “-” corresponds to an HCT116 cell lysate applied as a negative control. The expected molecular weights of the MCP-sfGFP and PCP-sfGFP proteins are 41.1 and 43.1 kDa, respectively, and these were consistent with the mobility of the proteins detected in the blot. (**B**) The distribution of HCT116_dCas9_MCP-sfGFP and HCT116_dCas9_PCP-sfGFP cells by sfGFP expression detected by flow cytometry. The vertical lines correspond to the medians of the distributions. (**C**) The qualitative analysis of the expression of sgRNAs. The electrophoresis of the products of the PCR amplification of cDNA from cells expressing the corresponding sgRNAs is shown. The sizes of several bands from the molecular length markers are indicated. Lane “HCT116” corresponds to HCT116 cells that did not express sgRNAs (negative control). The expected sizes of the PCR products were 107 bp for the sgRNA pair and 496 bp for the GAPDH pair. The expected sizes were consistent with the observed values. (**D**) The quantitative analysis of sgRNA expression by real-time PCR. Each column corresponds to a single cell culture expressing dCas9, MCP-sfGFP, or PCP-sfGFP and the indicated sgRNA(s). Values represent the mean ± SEM of the expression level relative to GAPDH gene expression for three independent RNA extractions. (**E**) *p*-values for the pairwise comparisons of the values shown in (**D**) using the Tukey HSD test. *p*-values lower than 0.05 are shown in bold. Values less than 1 × 10^−6^ are rounded to 0.0. The *p*-value for the preliminary one-way ANOVA was 2.9 × 10^−9^, allowing subsequent post hoc analysis by Tukey’s HSD test. (**F**) A scatter plot showing the values of the visualization efficiency against the relative expression of the sgRNAs shown in (**D**). No statistically significant correlation was found (Spearman correlation coefficient = 0.571, *p*-value = 0.139).

**Figure 6 cells-13-01440-f006:**
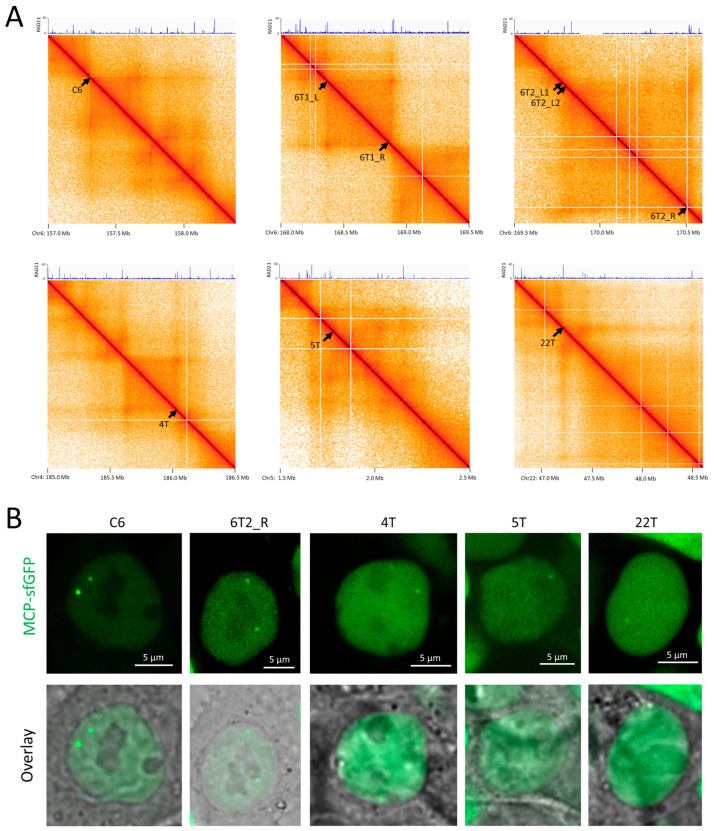
Visualization of the borders of topologically associating domains by CRISPR-Sirius. (**A**) Hi-C maps of the regions of designated chromosomes containing selected TAD borders (indicated by labeled arrows). ChIP-Seq profiles of fold enrichment for the RAD21 cohesin subunit are shown at the top of each map. The coordinates shown are for the hg38 human genome assembly. (**B**) Examples of microscope images of HCT116_dCas9_MCP-sfGFP cells transduced with visualizing gRNA genes for the indicated TAD borders.

**Figure 7 cells-13-01440-f007:**
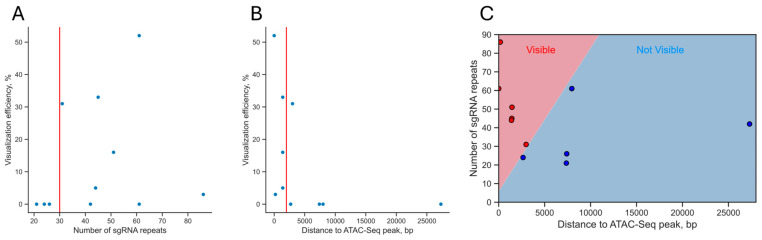
The relationship between the visualization efficiency of a locus and the number of sgRNA repeats and the distance to ATAC-Seq peaks. (**A**) Scatter plot showing the visualization efficiency against the number of sgRNA repeats in a locus. The vertical red line corresponds to 30 repeats per cluster (a possible lower limit of repeats in the visualizable clusters). (**B**) Scatter plot showing the visualization efficiency against the distance to the nearest ATAC-Seq peak. The vertical red line corresponds to 2000 bp (a possible upper limit of the distance for the visualizable clusters). (**C**) The distribution of the studied loci by the distance to the ATAC-Seq peak and by the number of sgRNA repeats in the locus. Red dots are loci that we were able to visualize using CRISPR-Sirius, and blue dots are those that we were not able to visualize. The boundary between regions corresponds to the decision border for logistic regression (*p*-value = 0.0005).

**Figure 8 cells-13-01440-f008:**
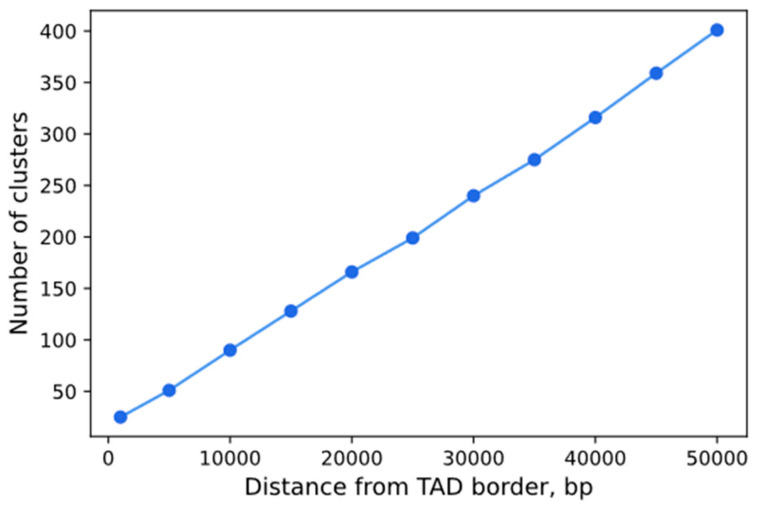
The number of clusters with locus-specific repeats that could potentially be visualized by the CRISPR-Sirius technology as a function of the distance to the TAD boundary across the genome in HCT116 cells. The list of eligible repeats was taken from genome.ucf.edu/CRISPRbar [22]. TAD boundaries in HCT116 cells were taken from Rao et al., 2017 [34]. Only clusters with at least one boundary within the specified distance from a TAD boundary were counted.

**Table 1 cells-13-01440-t001:** Primers used for molecular cloning.

Primer Name	Primer Sequence	Task
T2A_BamH_f	AATAAGGATCCGAGGGCAGAGGAAGTCTTCTAACAT	Replacing the P2A-HSA fragment in the dCas9 vector with the T2A-Puro fragment
Puro_Xba_r	AATAATCTAGATAGATCAGGCACCGGGCTT	
sfGFP_BamH_f	ATTAAGGATCCATGCGTAAAGGCGAAGAGCT	Replacing the HaloTag with sfGFP in a plasmid with the MCP gene
sfGFP_Xho_r	TAATTCTCGAGTTTGTACAGTTCATCCATACCATGCG	

**Table 2 cells-13-01440-t002:** sgRNA recognition sequences and the genomic coordinates of IDR3 and C6 loci.

Target Locus ^1^	sgRNA Recognition Sequence
IDR3 (chr19: 380,836–382,654)	AGCAGATGTAGG (45) ^2^
C6 (chr6: 157,310,367–157,314,361)	sg1: GTGAGTGCACAC (22) sg2: TGGGACACTATGATG (39)

^1^ Coordinates are for the hg38 human genome assembly. ^2^ Numbers in parentheses represent the number of sgRNA binding sites in the target loci.

**Table 3 cells-13-01440-t003:** Visualization efficiency for two types of CRISPR-Sirius system with different target loci.

Target	PCP-sfGFP Version		MCP-sfGFP Version	
	Transient Transfection	Stable Transduction	Transient Transfection	Stable Transduction
IDR3	6% (n = 130) ^1^	26% (n = 124)	13% (n = 245)	33% (n = 138)
C6	3% (n = 132)	20% (n = 102)	9% (n = 127)	52% (n = 131)

^1^ Values represent proportions of nuclei with at least one signal as a percentage of all nuclei (n: number of nuclei analyzed).

**Table 4 cells-13-01440-t004:** sgRNA recognition sequences and the genomic coordinates of chosen TAD borders.

Target Locus ^1^	sgRNA Recognition Sequence
C6 (chr6: 157,310,367–157,314,361)	sg1: GTGAGTGCACAC (22) ^2^ sg2: TGGGACACTATGATG (39)
6T1_L (chr6: 168,378,356–168,380,872)	sg1: ACTCGGGCTGTG (35) sg2: CTGTGTGGGACT (26)
6T1_R (chr6: 168,849,859–168,850,601)	sg1: GCAGAGGTGGCA (22) sg2: TGTGGGCAGAGG (20)
6T2_L (chr6: 169,781,629–169,782,955 for sg1, and chr6: 169,803,533–169,807,849 for sg2)	sg1: ACCACTCGGAAA (21) sg2: GCTCTGTGTCTG (24)
6T2_R (chr6: 170,500,882–170,504,181 for sg1, and chr6: 170,507,778–170,509,799 for sg2)	sg1: CTGCAGCCATCA (31) sg2: CACTCATTCAGC (26)
4T (chr4: 186,033,845–186,035,464)	sg1: CCCTGAGGGATT (22) sg2: TCTGTACCCTGA (29)
5T (chr5: 1,781,394–1,781,810)	sg1: AGGCTGAGGGTG (21) sg2: AGGGTGAGGCTG (23)
22T (chr22: 47,211,081–47,213,579)	sg1: CATATTTGAGTG (56) sg2: GGACGGTCAGTG (30)

^1^ Coordinates are for the hg38 human genome assembly. ^2^ Numbers in parentheses represent the number of sgRNA-binding sites in the target loci.

**Table 5 cells-13-01440-t005:** Visualization efficiency of two versions of the CRISPR-Sirius system for different target TAD borders.

Target	PCP-sfGFP Version	MCP-sfGFP Version	Proportion of Cells with Two Signals ^2^	Signal-to-Background Ratio (Median)
C6	20% (n = 102) ^1^	52% (n = 131)	34%	2.3
6T1_L	0% (n = 118)	0% (n = 113)	0%	n.a.
6T1_R	0% (n = 121)	0% (n = 166)	0%	n.a.
6T2_L	0% (n = 108)	0% (n = 122)	0%	n.a.
6T2_R	0% (n = 134)	29% (n = 159)	8%	1.5
4T	0% (n = 127)	16% (n = 117)	4%	1.4
5T	0% (n = 123)	5% (n = 129)	0%	1.5
22T	0% (n = 158)	3% (n = 116)	0%	1.4

^1^ Values represent proportions of nuclei with at least one signal as a percentage of all nuclei (n: number of nuclei analyzed). ^2^ Of the overall number of nuclei analyzed.

**Table 6 cells-13-01440-t006:** Properties of the target repeat clusters.

Repeat Cluster	Visualization Efficiency (MCP-sfGFP Version)	Number of sgRNA Repeats	Chromatin Status (ChromHMM18)	Transcription	Hi-C Compartment
IDR3	33% ^1^	45	Quiescent	No	A
C6	52%	61	Quiescent	Weak	A
6T1_L	0%	61	Quiescent	No	A
6T1_R	0%	42	Quiescent	No	A
6T2_L_sg1	0%	21	Quiescent	No	A
6T2_L_sg2	0%	24	Repressed Polycomb/weakly repressed Polycomb	No	A
6T2_R_sg1	31%	31	Weakly Polycomb repressed	No	A
6T2_R_sg2	0%	26	Quiescent	No	A
4T	16%	51	Quiescent	No	A
5T	5%	44	Quiescent	No	A
22T	3%	86	Quiescent	No	B

^1^ Values represent the proportions of nuclei with at least one signal as a percentage of all nuclei.

## Data Availability

Data are contained within the article and Appendix A.

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
