# Peer review of "A Comparison of Two Versions of the CRISPR-Sirius System for the Live-Cell Visualization of the Borders of Topologically Associating Domains"

_cells, 2024, doi:10.3390/cells13171440_

Round 1

Reviewer 1 Report

Comments and Suggestions for Authors

This is a technical report that will be much appreciated by the community of researchers looking to use CRISPR-dCas9-related technologies to visualize chromatin structure and dynamics by microscopy. The authors performed a well-controlled comparison of two RNA-binding proteins, MS2-binding MCP and PP7-binding PCP, as well as surveying different genomic loci corresponding to TAD boundaries. The results indicate that MCP is a better choice for most loci, with a higher efficiency of visualization. The authors also confirmed that lentiviral delivery of gRNAs yields a higher % of cells showing the desired fluorescent spots in the nuclei.

First, I have a minor suggestion: To improve the interpretation of the variable visualization efficiency across the several genomic loci, shown in Table 5, why not add a column showing the number of repeats? The authors discuss that the number of repeats does not explain the widely different efficiency in the main text, but it will be easier to see that observation if shown as part of the table.

Second, have the authors considered epigenetic features (other than cohesion ChIP-seq peaks) that might partly explain the variable visualization efficiency? On a global scale, it would be of interest to indicate which of the chosen loci are in compartment A versus B (obtained from the Hi-C data). On a finer scale, are some of the loci in heterochromatin regions, e.g. based on repressive histone mark ChIP-seq or lack of ATAC-seq peaks? If a chosen locus is in highly condensed repressive chromatin, then gRNAs may not access the site for specific visualization. Since HCT116 is a commonly used cell line, some epigenomic data are likely available in the public databases for the authors to assess the overlap between these chromatin features and the chosen loci. This additional information might possibly reveal new patterns that partially explain the variable efficiency.

Author Response

Comment 1: This is a technical report that will be much appreciated by the community of researchers looking to use CRISPR-dCas9-related technologies to visualize chromatin structure and dynamics by microscopy. The authors performed a well-controlled comparison of two RNA-binding proteins, MS2-binding MCP and PP7-binding PCP, as well as surveying different genomic loci corresponding to TAD boundaries. The results indicate that MCP is a better choice for most loci, with a higher efficiency of visualization. The authors also confirmed that lentiviral delivery of gRNAs yields a higher % of cells showing the desired fluorescent spots in the nuclei.”

Response 1: We are thankful for the thorough review of our manuscript.

Comment 2: First, I have a minor suggestion: To improve the interpretation of the variable visualization efficiency across the several genomic loci, shown in Table 5, why not add a column showing the number of repeats? The authors discuss that the number of repeats does not explain the widely different efficiency in the main text, but it will be easier to see that observation if shown as part of the table.

Response 2: Thank you for your suggestion, this will indeed make our conclusions more substantiated. We have added a new Section 3.6 “Analysis of the dependence of visualization efficiency on the number of sgRNA repeats in a cluster and epigenetic factors”  that considers the relationship between visualization efficiency and the number of target repeats in the clusters and the epigenetic factors. This Section is equipped with Figure 7 and Table 6. Particularly, Figure 7A and the Column “Number of sgRNA repeats” of Table 6 contain the relevant information.

Comment 3: "Second, have the authors considered epigenetic features (other than cohesion ChIP-seq peaks) that might partly explain the variable visualization efficiency? On a global scale, it would be of interest to indicate which of the chosen loci are in compartment A versus B (obtained from the Hi-C data). On a finer scale, are some of the loci in heterochromatin regions, e.g. based on repressive histone mark ChIP-seq or lack of ATAC-seq peaks? If a chosen locus is in highly condensed repressive chromatin, then gRNAs may not access the site for specific visualization. Since HCT116 is a commonly used cell line, some epigenomic data are likely available in the public databases for the authors to assess the overlap between these chromatin features and the chosen loci. This additional information might possibly reveal new patterns that partially explain the variable efficiency."

Response 3: Indeed, it would be interesting to look for possible reasons for the different efficiency of visualization using the CRISPR-Sirius system, based on the available epigenetic data. As we mentioned in a previous answer, we added a section 3.6, containing the information about the vicinity of target repeat clusters to the ATAC-Seq peaks, and their belongingness to A or B compartments. Since  separate histone mark tracks may be insufficient to draw conclusions due to the interplay between many different epigenetic marks, we added information about chromatin types to which the studied loci belong according to ChromHMM 18-state chromatin model. This model considers several histone modifications and therefore provides an integral picture of the chromatin status (Ernst and Kellis, 2012, Ernst and Kellis, 2017). The chromatin state of each locus can be found in Table 6 and is also discussed in Section 3.6 of the revised manuscript. We have added separate histone marking profiles, as well as ATAC-Seq profile, PolyA-plus RNA-seq profile and ChromHMM18 track for each locus as Supplementary figures S3-S11. The links to the studied epigenetic datasets are provided in “Epigenetic data''  subsection of  “Materials and Methods”. 

We realize, however, that the diversity of the loci we studied is limited, and therefore it is difficult to talk about the biological significance of the patterns. They are presented rather from a descriptive point of view, but where possible, we added an assessment of the statistical significance of the findings.

References: 

Ernst, J.; Kellis, M. ChromHMM: automating chromatin-state discovery and characterization. Nat Methods 2012, 9, 215-216, doi:10.1038/nmeth.1906.

Ernst, J.; Kellis, M. Chromatin-state discovery and genome annotation with ChromHMM. Nat Protoc 2017, 12, 2478-2492, doi:10.1038/nprot.2017.124.

Reviewer 2 Report

Comments and Suggestions for Authors

In this manuscript entitled “A comparison of two variants of the CRISPR-Sirius system for live-cell visualization of the borders of topologically associated domains, Viushkov et al compared the efficiency of two delivery methods (Transient transfection with Lipofectamine 3000 and cell transduction) of the guide RNA in the case of CRISPR-Sirius and concluded that cell transduction gave a higher proportion of cells with 1 or 2 signals. Also, they compared two systems : PCP-sfGFP with the sgRNA with 8xPP7 repeats or MCP-sfGFP with the sgRNA with 8xMS2 repeats, for their efficiency in visualizing 8 loci located at a TAD border in HCT116 cells and observed that, in their hands, the MS2-MCP system was more efficient than the PP7-PCP system and gave a higher proportion of cells with at least one signal.

Comments on part 3.1. and 3.2

The fact that cell transduction is more efficient with the Crispr system than transient transfection was expected and most of the labs doing Crispr generally use cell transduction.

However, the very low efficiency observed by the authors with the PP7-PCP system is surprising as previous publications have not reported such a difference between the two systems.

Could the authors state clearly the differences between their system and the system used by other groups (see for example the papers from Ma et al (2018) that the authors reference) ?

The authors should mention if they are using the sgRNA with the repeats at the 3’end or at the tetraloop of the guide RNA. It has been shown by Ma et al. (2018) that having the repeats at the tetraloop gave much better results.

In addition, some more experiments should be done in order to understand what causes the low efficiency of their PP7-PCP-system.

The level of PCP-sfGFP and MCP-sfGFP should be tested, for example by western blot, to make sure that the 2 fusion proteins are properly expressed and stable.

The control of the expression and stability of the sgRNAs by qPCR should be performed.

Also, it is not clear why the authors are using 2 sgRNAs for most of the loci? It would be interesting to test each sgRNA individually, check their stability and their efficiency and then compare the results with the 2 used in combination.

Minor comment

A schematic representation of the constructs used in each system would be useful.

The authors should replace the word “variant”, which could be misleading, by “system” or “version”.

Comments on Part 3.3.

Could the authors be precise about how many loci, out of the 1200 loci with more than 20 repeats, are at a TAD border in the HCT116 cells This would give an idea of the limit of the system to study chromatin architecture.

The authors should state the proportion of cells with 2 signals and the signal to noise ratio to really evaluate the efficiency of the system for each locus.

Finally, visualising a locus located at a TAD border in one particular cell type does not give any information on chromatin organisation. The interesting point in this case is to look at the distance between the 2 loci that are flanking a TAD and monitor their distance to see if the TAD is formed or not under a specific condition. Of course, this require the use of 2 distinct fluorescent proteins, which, in the case of this study, has not been performed and would be difficult considering the low efficiency of the PP7-PCP system.

Author Response

Comment 1:  “In this manuscript entitled “A comparison of two variants of the CRISPR-Sirius system for live-cell visualization of the borders of topologically associated domains, Viushkov et al compared the efficiency of two delivery methods (Transient transfection with Lipofectamine 3000 and cell transduction) of the guide RNA in the case of CRISPR-Sirius and concluded that cell transduction gave a higher proportion of cells with 1 or 2 signals. Also, they compared two systems : PCP-sfGFP with the sgRNA with 8xPP7 repeats or MCP-sfGFP with the sgRNA with 8xMS2 repeats, for their efficiency in visualizing 8 loci located at a TAD border in HCT116 cells and observed that, in their hands, the MS2-MCP system was more efficient than the PP7-PCP system and gave a higher proportion of cells with at least one signal.”

Response 1:  We are thankful for the thorough review of our manuscript.

Comment 2: Comments on part 3.1. and 3.2. The fact that cell transduction is more efficient with the Crispr system than transient transfection was expected and most of the labs doing Crispr generally use cell transduction.”

Response 2: Indeed, given that the transduction efficiency should be higher on average than that of transient transfection, one would expect that the imaging efficiency in case of guide RNA gene delivery by transduction should also be higher than that of transfection. However, many studies on CRISPR imaging used transient plasmid transfection to deliver guide RNA genes and achieved high imaging efficiencies (Ma et al., 2016; Hong et al. 2016; Qin et al., 2017; Chaudhary 2020; Clow et al., 2022). The original CRISPR-Sirius paper used both Lipofectamine transfection and lentiviral transduction to deliver guide RNA genes, but did not explicitly compare the efficiencies of the two approaches (Ma et al., 2018). It could be expected that with high transfection efficiency it would be possible to achieve a higher number of plasmid copies, and thus a higher copy number of the guide RNA gene in the cell, than as a result of transduction and integration of genes into the genome. Considering also that transfection is a less labor-intensive method, since it does not require preliminary production of lentiviruses, we decided to compare two methods of delivering guide RNA genes of the CRISPR-Sirius system under our conditions. Our results indicate that transduction followed by selection of cells on an antibiotic allows achieving higher visualization efficiency, and therefore we recommend giving preference to this method of delivering guide RNA genes.

References:

Ma, H.; Tu, L.C.; Naseri, A.; Huisman, M.; Zhang, S.; Grunwald, D.; Pederson, T. Multiplexed labeling of genomic loci with dCas9 and engineered sgRNAs using CRISPRainbow. Nat Biotechnol 2016, 34, 528-530, doi:10.1038/nbt.3526. 

Ma, H.; Tu, L.C.; Naseri, A.; Chung, Y.C.; Grunwald, D.; Zhang, S.; Pederson, T. CRISPR-Sirius: RNA scaffolds for signal amplification in genome imaging. Nat Methods 2018, 15, 928-931, doi:10.1038/s41592-018-0174-0.

Chaudhary, N.; Nho, S.H.; Cho, H.; Gantumur, N.; Ra, J.S.; Myung, K.; Kim, H. Background-suppressed live visualization of genomic loci with an improved CRISPR system based on a split fluorophore. Genome Res 2020, 30, 1306-1316, doi:10.1101/gr.260018.119.

Clow, P.A.; Du, M.; Jillette, N.; Taghbalout, A.; Zhu, J.J.; Cheng, A.W. CRISPR-mediated multiplexed live cell imaging of nonrepetitive genomic loci with one guide RNA per locus. Nat Commun 2022, 13, 1871, doi:10.1038/s41467-022-29343-z.

Hong, Y.; Lu, G.; Duan, J.; Liu, W.; Zhang, Y. Comparison and optimization of CRISPR/dCas9/gRNA genome-labeling systems for live cell imaging. Genome Biol 2018, 19, 39, doi:10.1186/s13059-018-1413-5.

Qin, P.; Parlak, M.; Kuscu, C.; Bandaria, J.; Mir, M.; Szlachta, K.; Singh, R.; Darzacq, X.; Yildiz, A.; Adli, M. Live cell imaging of low- and non-repetitive chromosome loci using CRISPR-Cas9. Nat Commun 2017, 8, 14725, doi:10.1038/ncomms14725.

Comment 3: However, the very low efficiency observed by the authors with the PP7-PCP system is surprising as previous publications have not reported such a difference between the two systems.”

Response 3: We were also surprised by such a big difference. But it is worth noting that the CRISPR-Sirius system does not use the basic, widely used versions of the MS2 and PP7 stem-loops, but rather their modified variants optimized by the authors of this method specifically for visualization purposes. We carefully reviewed all three articles that use the CRISPR-Sirius technology (Ma et al., 2018, Ma et al., 2019, Chung et al., 2023) and did not find a direct comparison of the efficiency of the two variants of this system (MS2/MCP and PP7/PCP). Our results indicate that the MS2/PCP variant is more efficient. 

References:

Ma, H.; Tu, L.C.; Naseri, A.; Chung, Y.C.; Grunwald, D.; Zhang, S.; Pederson, T. CRISPR-Sirius: RNA scaffolds for signal amplification in genome imaging. Nat Methods 2018, 15, 928-931, doi:10.1038/s41592-018-0174-0.

Ma, H.; Tu, L.C.; Chung, Y.C.; Naseri, A.; Grunwald, D.; Zhang, S.; Pederson, T. Cell cycle- and genomic distance-dependent dynamics of a discrete chromosomal region. J Cell Biol 2019, 218, 1467-1477, doi:10.1083/jcb.201807162.

Chung, Y.C.; Bisht, M.; Thuma, J.; Tu, L.C. Single-chromosome dynamics reveals locus-dependent dynamics and chromosome territory orientation. J Cell Sci 2023, 136, doi:10.1242/jcs.260137.

Comment 4: “Could the authors state clearly the differences between their system and the system used by other groups (see for example the papers from Ma et al (2018) that the authors reference) ?”

Response 4: We use the same system (CRISPR-Sirius) proposed by Ma et al., 2018 (and then used in two other papers - Ma et al., 2019 and Chung et al., 2022). We described the principle of the CRISPR-Sirius system, providing links to the original papers, in lines 53-71 of the original submitted manuscript (this information remained in the revised manuscript, lines 53-72). We explicitly indicated that we use this particular system in lines 73-75 of the original submitted manuscript (this information remained in the revised manuscript, lines 82-84).

Our modifications of the CRISPR-Sirius system (changing the selectable marker in dCas9 to puromycin resistance ORF, as well as replacing the HaloTag fluorescent protein in MCP with sgGFP) are described in the “Plasmid construction” section of the “Materials and methods”, as well as in the Results section (lines 309-314, and 369-371). Our modifications did not concern the sequences of the guide RNAs with PP7 or MS2 stem-loops, or the sequences of the MCP and PCP proteins. These proteins and stem-loops are the same as in the original version of the CRISPR-Sirius technology (Ma et al., 2018).

(References are provided in the previous response)

Comment 5: “The authors should mention if they are using the sgRNA with the repeats at the 3’end or at the tetraloop of the guide RNA. It has been shown by Ma et al. (2018) that having the repeats at the tetraloop gave much better results.”

Response 5: In CRISPR-Sirius technology MS2 or PP7 repeats are added to the tetraloop, and this is the technology we used. For clarity, we added the information that MS2/PP7 stem-loops are inserted into the tetraloop in the text of the revised manuscript (line 54), and also added Figure 1 illustrating the organization of the CRISPR-Sirius system.

Comment 6: In addition, some more experiments should be done in order to understand what causes the low efficiency of their PP7-PCP-system.

The level of PCP-sfGFP and MCP-sfGFP should be tested, for example by western blot, to make sure that the 2 fusion proteins are properly expressed and stable.

The control of the expression and stability of the sgRNAs by qPCR should be performed.”

Response 6: Thank you for this suggestion. We agree, that the underlying differences in the efficiency of the two versions of the CRISPR-Sirius can be further elucidated, and that measuring the expression of PCP-sfGFP and MCP-sfGFP proteins as well as corresponding sgRNAs can provide valuable clues. 

We added Section 3.4 named “Analysis of the expression of sgRNAs and stem-loop binding proteins in two versions of the CRISPR-Sirius system.” that contains the results of the suggested tests. This section is provided with Figure 5. The utilized protocols for Western blot and real-time PCR were added to the “Materials and Methods” section.

Comment 7: “Also, it is not clear why the authors are using 2 sgRNAs for most of the loci? It would be interesting to test each sgRNA individually, check their stability and their efficiency and then compare the results with the 2 used in combination.

Response 7: Thank you for pointing this out. We suggested that by using two sgRNA we will recruit more fluorescent proteins to a locus of interest. We added Section 3.3 named  “Evaluation of the imaging performance using a single guide RNA per locus” that compares the efficiency of using single or paired sgRNAs per locus. The results of expression assessment for single or paired sgRNAs are provided in section 3.4 in the context of the comparison of MS2/MCP and PP7/PCP versions of the CRISPR-Sirius system.

Comment 8: "Minor comment. A schematic representation of the constructs used in each system would be useful.”

Response 8: Thank you for this suggestion. We added a schematic representation of the CRISPR-Sirius system as Figure 1.

Comment 9: "The authors should replace the word “variant”, which could be misleading, by “system” or “version”.

Response 9: We replaced the word “variant” with “version” in the manuscript title, as well as everywhere else it appeared in the text in this context (with “version” or “type”).

Comment 10:  “Comments on Part 3.3. Could the authors be precise about how many loci, out of the 1200 loci with more than 20 repeats, are at a TAD border in the HCT116 cells This would give an idea of the limit of the system to study chromatin architecture.”

Response 10: Indeed, this information would be valuable for the paper. However, the concept of proximity is debatable. What should be the maximum distance of a repeat cluster from a TAD boundary to consider that we are visualizing a TAD boundary? Should this distance not exceed 1000 bp, or maybe 10 or even 50 kbp? The answer to this question, in our opinion, depends on the biological context of a particular study. From the point of view of optical microscopy, all these distances will be localized at approximately the same point due to the limitation of the resolution of optical microscopy. In order not to limit the reader to a specific fixed distance, we have added to the discussion a graph that reflects the number of repeat clusters at a given distance from TAD boundaries throughout the genome (Figure 8).

Comment 11: "The authors should state the proportion of cells with 2 signals and the signal to noise ratio to really evaluate the efficiency of the system for each locus"

Response 11:  We have added this information to Table 5.

Comment 12: “Finally, visualising a locus located at a TAD border in one particular cell type does not give any information on chromatin organisation. The interesting point in this case is to look at the distance between the 2 loci that are flanking a TAD and monitor their distance to see if the TAD is formed or not under a specific condition. Of course, this require the use of 2 distinct fluorescent proteins, which, in the case of this study, has not been performed and would be difficult considering the low efficiency of the PP7-PCP system."

Response 12: Yes, we agree. We worked only in one fluorescence channel, and therefore we cannot visualize the two boundaries of the TAD, which would be of great interest. Nevertheless, we hope that the results of our study, and especially the set of TAD boundaries, will be useful in possible future works dedicated to multichannel systems.

Round 2

Reviewer 1 Report

Comments and Suggestions for Authors

The authors have addressed my previous comments well in the revised manuscript.

Author Response

Comment 1: "The authors have addressed my previous comments well in the revised manuscript."

Response 1: "We are glad that you are satisfied with the revised version of the manuscript."

Reviewer 2 Report

Comments and Suggestions for Authors

I thank the authors for their answers to my comments and for performing supplementary experiments to help understand the difference observed between the 2 systems (PCP-sfGFP and MS2-sfGFP) studied in this manuscript.

Also, I appreciate the effort made to correlate the visualisation efficiency with the epigenetic context even if, as stated by the authors, the number of loci tested is too low to draw definitive conclusions.  I find the new version of the manuscript interesting and I have no further major issues with it.

Minor comment : 

Table 5 :  Footnote 3 mentioned that the overall number of repeats for 2 sgRNA is displayed in the table but I cannot find the information in this table. The authors should either remove footnote 3 or add the information in the table.  

Author Response

Comment 1: "Table 5 :  Footnote 3 mentioned that the overall number of repeats for 2 sgRNA is displayed in the table but I cannot find the information in this table. The authors should either remove footnote 3 or add the information in the table. "

Response 1: "We are glad that you are satisfied with the revised version of the manuscript. The Footnote 3 was left under Table 5 by mistake and is indeed irrelevant to this table. We have removed this typo. Thank you for your careful reading of our manuscript."